

# Identification of olfactory genes of a forensically important blow fly, *Aldrichina grahami* (Diptera: Calliphoridae)

Han Han[1,*], Zhuoying Liu[1,2,*], Fanming Meng[1], Yangshuai Jiang[1] and Jifeng Cai[1]

[1] Department of Forensic Science, School of Basic Medical Sciences, Central South University, Changsha, Hunan, China
[2] Departments of Anesthesiology and Medicine, David Geffen School of Medicine at University of California Los Angeles, Los Angeles, CA, USA
* These authors contributed equally to this work.

Corresponding authors
Fanming Meng,
mengfanming1984@163.com
Jifeng Cai, cjf_jifeng@163.com

## ABSTRACT

**Background:** The time-length between the first colonization of necrophagous insect on the corpse and the beginning of investigation represents the most important forensic concept of minimum post-mortem inference (PMImin). Before colonization, the time spent by an insect to detect and locate a corpse could significantly influence the PMImin estimation. The olfactory system plays an important role in insect food foraging behavior. Proteins like odorant binding proteins (OBPs), chemosensory proteins (CSPs), odorant receptors (ORs), ionotropic receptors (IRs) and sensory neuron membrane proteins (SNMPs) represent the most important parts of this system. Exploration of the above genes and their necrophagous products should facilitate not only the understanding of their roles in forging but also their influence on the period before PMImin. Transcriptome sequencing has been wildly utilized to reveal the expression of particular genes under different temporal and spatial condition in a high throughput way. In this study, transcriptomic study was implemented on antennae of adult *Aldrichina grahami* (Aldrich) (Diptera: Calliphoridae), a necrophagous insect with forensic significance, to reveal the composition and expression feature of OBPs, CSPs, ORs, IRs and SNMPs genes at transcriptome level.
**Method:** Antennae transcriptome sequencing of *A. grahami* was performed using next-generation deep sequencing on the platform of BGISEQ-500. The raw data were deposited into NCBI (PRJNA513084). All the transcripts were functionally annotated using Gene Ontology (GO) and the Kyoto Encyclopedia of Genes and Genomes (KEGG) database. Differentially expressed genes (DEGs) were analyzed between female and male antennae. The transcripts of OBPs, CSPs, ORs, IRs and SNMPs were identified based on sequence feature. Phylogenetic development of olfactory genes of *A. grahami* with other species was analyzed using MEGA 5.0. RT-qPCR was utilized to verify gene expression generated from the transcriptome sequencing.
**Results:** In total, 14,193 genes were annotated in the antennae transcriptome based on the GO and the KEGG databases. We found that 740 DEGs were differently expressed between female and male antennae. Among those, 195 transcripts were annotated as candidate olfactory genes then checked by sequence feature. Of these,

27 OBPs, one CSPs, 49 ORs, six IRs and two SNMPs were finally identified in antennae of *A. grahami*. Phylogenetic development suggested that some olfactory genes may play a role in food forging, perception of pheromone and decomposing odors.

**Conclusion:** Overall, our results suggest the existence of gender and spatial expression differences in olfactory genes from antennae of *A. grahami*. Such differences are likely to greatly influence insect behavior around a corpse. In addition, candidate olfactory genes with predicted function provide valuable information for further studies of the molecular mechanisms of olfactory detection of forensically important fly species and thus deepen our understanding of the period before PMImin.

## INTRODUCTION

Forensic entomology uses insects to help in determining origin, location and time of death of a human. The estimation of the postmortem interval by forensic entomologists is based on the development of the insects that colonize the corpse. The time elapsed between the first insect colonizing on the corpse to the start of forensic examination represents the most important forensic concept of minimum post-mortem inference (PMImin). When and how the first insect detects and colonizes the corpse could obviously affect the beginning of the PMImin. The length of time between the time point of death and the first insect colonization on corpse was defined as the pre-colonization interval (Pre-CI), which would account for large part of decomposition time depending on various conditions (*Tomberlin et al., 2011*). Thus, the PMImin error could vary from a few hours to a few days when pre-CI was overlooked unconsciously, which could limit the application of forensic entomology. Some irritant compounds, such as volatile compounds (VOCs), are produced during the process of corpse decomposition (*Rosier et al., 2016*). It is believed that these VOCs provide clues for host detection and oviposition in necrophagous flies. Studies have shown that these behaviors can be mediated by the olfactory system and may exhibit gender difference (*Wicker-Thomas, 2007*).

The insects' olfactory system is a highly specific and extremely sensitive chemical sensory nervous system, formed during long-term evolution. Antennae are the main olfactory organs in insects. They have critical roles in detecting environmental chemical signals and subsequently affecting insect behaviors (*Zhou et al., 2015*; *Das et al., 2011*), such as mate choice, host searching, oviposition site selection and toxic compound avoidance (*Liu et al., 2018*). Olfactory proteins are commonly expressed in olfactory tissue antennae and maxillary palp, or in non-olfactory like tentacles, lower lip whiskers, and so on (*De Bruyne, Clyne & Carlson, 1999*; *Kwon et al., 2006*). One study suggests that different types of olfactory-related proteins could participate in sensing different odors (*Hallem & Carlson, 2006*). Generally, olfactory-related proteins are classified as odorant

binding proteins (OBPs), chemosensory proteins (CSPs), odorant receptors (ORs), ionotropic receptors (IRs), sensory neuron membrane proteins (SNMPs) and odor degrading enzymes (ODEs) (*Gu et al., 2015*; *Leal, 2013*; *Wang, Liu & Wang, 2017*).

The OBPs and the CSPs represent the key step in the insect olfactory signaling process (*Zhou et al., 2010*). They are small soluble proteins in the sensilla lymph, and when combined with chemical molecules form a conjugate which could be recognized by corresponding receptors (*Gong et al., 2007*; *Pelosi et al., 2006*). The OBPs are highly conserved protein, firstly discovered in *Antheraea polyphemus* (*Vogt & Riddiford, 1981*). Notably, many OBPs have binding preference or higher affinity to specific odor compounds (*Maida, Ziegelberger & Kaissling, 2003*; *Pophof, 2004*). Based on the literature, OBPs can be divided into four sub-types according to sequence feature: "Classical" OBPs, "Dimer" OBPs, "Minus-C" OBPs and "Plus-C" OBPs (*Campanini & De Brito, 2016*; *Zhou et al., 2004*). In 2011, there is a study further classified the classical OBPs into three sub-families, as pheromone-binding proteins (PBPs), general odorant-binding proteins (GOBPs) and antennal-binding protein X homologs (ABPXs) (*Zhang et al., 2011*). Compared OBPs, CSPs are mainly involved in insects' perception of chemical signals and exhibit multifarious functions (*Jacquin-Joly et al., 2001*; *Mei et al., 2018*).

The ORs are participants in chemosensory signal transduction processes. ORs have seven transmembrane domains and a specific reversed membrane topology (*Benton, 2006*; *Jacquin-Joly & Merlin, 2004*; *Clyne et al., 1999*). In insects' olfactory nervous systems, there are two types of ORs: Conventional ORs and olfactory receptor co-receptor (ORCo), previously known as OR83b (*Larsson et al., 2004*; *Smith, 2007*). The conventional ORs, a highly divergent family, respond to pheromones and VOCs (*Carey et al., 2010*; *Mitchell et al., 2012*; *Vosshall, Wong & Axel, 2000*). Compared with traditional ORs, ORCo is a highly-conserved family (*Jones et al., 2005*). They act as ion channel (*Mitchell et al., 2012*; *Sato et al., 2008*), but their involvement in olfactory transduction remains controversial (*Zufall & Domingos, 2018*).

The IRs are also membrane-bound chemosensory receptors located in the dendritic membrane of receptor neurons like ORs (*Benton et al., 2009*). Based on their function, IRs are divided into two types: the conserved "antennal IRs" and the species-specific "divergent IRs" (*Benton et al., 2009*; *Rimal & Lee, 2018*; *Wang et al., 2016*). The antennal IRs are mainly involved in the sensory process of odorant, while divergent IRs plays a role in taste (*Benton et al., 2009*; *Croset et al., 2010*; *Wang et al., 2016*).

Likewise, the SNMPs are transmembrane proteins, homologous to the mammalian CD36 protein family (*Nichols & Vogt, 2008*; *Rogers et al., 1997*), and are important for odor detection (*Hu et al., 2016*). Subtypes SNMP1 and SNMP2 are organs specifically expressed at different locations (*Rogers, Krieger & Vogt, 2001*; *Rogers, Steinbrecht & Vogt, 2001*; *Rogers et al., 1997*). Indeed, the SNMP1 is especially expressed in the dendritic membrane of olfactory receptor neurons, and functions as a pheromone induction (*Rogers, Steinbrecht & Vogt, 2001*; *Vogt et al., 2009*; *Zhang et al., 2015a*), whereas the SNMP2 is found in supporting cell (*Forstner et al., 2008*; *Liu et al., 2013*; *Zhang et al., 2015a*), but its function is uncertain.

*Aldrichina grahami* (Aldrich) (Diptera: Calliphoridae) is a common necrophagous insect of forensic importance. It feeds on corpses or feces and mainly distributes in the Palearctic and the partial Oriental regions. Research reported its intrusion into other parts of the world in the past decades (*Sukontason et al., 2004*; *Fan, 1992*). *A. grahami* is a species with characteristic strong cold tolerance (*Wang et al., 2018*), and it could be the first and major insect species which locate and colonize on corpses in early spring and late autumn when temperature is relative low (*Guo et al., 2011*; *Kurahashi et al., 1984*). This means seasonal distribution of *A. grahami* could be useful for the PMI estimation in cold environment. Moreover, the emission of VOCs from corpses could be reduce under the low temperature (*Forbes et al., 2014*). In addition, *A. grahami* should be able to detect VOCs of corpses even when it at relative low density. The report of myiasis by *A. grahami* shows that the olfactory system is also helpful in host detection (*Liu, 1980*; *Yinong Duan & Zhou, 2002*). However, so far, there has been no study about the olfactory system of *A. grahami*. Considering that olfactory related proteins play a critical role in olfactory system, exploration of their genes and products should deepen our understanding of its behavior like food forging, locating, host recognizing and colonization as well as further improve methods for an accurate PMI estimation.

In this study, the first antennal transcriptome analysis of *A. grahami* was performed using next generation sequencing (NGS) to identify the genes of olfactory family from *A. grahami*. Differences between female and male gene expression were analyzed. A set of putative OBPs, CSPs, ORs, IRs and SNMP in *A. grahami* was annotated and the expression level was verified. Additionally, different expression profiles of the olfactory related proteins between female and male organs were explored. Finally, predicting function of olfactory genes was discussed based on phylogenetic analysis.

## MATERIALS AND METHODS

### Insect rearing

The first generation of *A. grahami* was captured using pork as baits in Changsha, Hunan province, China. Adults were identified based on the description of Fan (*Fan, 1992*). Both sexes were bred in plastic containers (30 cm × 30 cm × 30 cm) and reared with milk and sugar (1:1) as food sources to gain more individuals for sampling. The daylight regime was 12:12 (L:D) and the temperature in the rearing room was 25 + 2 °C with 70–80% relative humidity. Adult antennae were obtained from females and males under a dissecting microscope and flash-frozen in liquid nitrogen in 1.5 mL microcentrifuge tubes and then stored at −80 °C until used to isolate RNA.

### RNA isolation and quality assessment

Total RNA samples of antennae were isolated using TRIzol Reagent according to the manufacturer's protocol (invitrogen, Carlsbad, CA, USA). The quality of RNA was confirmed using a NanoVue UV–Vis spectrophotometer (GE Healthcare Bio-Science, Sweden, Europe), and RNA integrity was verified using a standard 1% agarose gel electrophoresis. Purification of RNA was carried out using DNase I as per manufacturer's instructions (Takara, Tokyo, Japan).

## CDNA library preparation and sequencing

Total RNA was treated by enriching poly-A tail mRNA with magnetic beads with OligodT, and the desired RNA was obtained after purification. Subsequently, the RNA was fragmented with a break buffer, and reverse transcribed with random N6 primers to synthesize cDNA double strands to obtain double stranded DNA. The synthesized double-stranded DNA was flattened at the end and phosphorylated at the 5′ end, flattened at the 3′ end with sticky' A′, and connected with adaptor. The ligation product was amplified by two specific primers then denatured by heat. Single-stranded DNA was cyclized with a bridge primer to obtain a single-stranded circular DNA library. Finally, the cDNA libraries were sequenced on the BGISEQ-500 sequencing platform (BGI-Shenzhen, Guangdong, China).

## Sequence reads mapping, assembly and annotation

Primarily, the raw reads were filtered by removing reads that contain adapters, poly nitrogen and low quality. The remaining high quality clean reads had a base quality 20% lower than Q20. Consequently, we calculated the Q20, Q30, GC-content and sequence duplication levels of the clean data. All subsequent analyses were performed using high quality clean reads. Clean reads were mapped to the *A. grahami* genome assembly (NCBI: PRJNA513084) by using HISAT2. (Parameter: —dta —phred64 unstranded —new-summary -x index -1 read_r1 -2 read_r2 (PE).)

## Analysis of differentially expressed genes

The quantity of gene expression levels from male and female groups were performed using FPKM (Fragments per kilobase of transcript per million mapped reads). It was calculated with NCBI gtf file through gene length annotation. The count calculation was performed using the HTSeq (*Anders, Pyl & Huber, 2015*). The Differentially expressed genes (DEGs) between male and female groups were identified by the DEG-seq (an R package to identify DEGs from RNA-Seq data) (*Wang et al., 2010*). To improve the accuracy of the DEGs, the DEGs were filtered with a fold change >2 or <0.5, and the false discovery rates (FDR) <0.05 (*Anders & Huber, 2010*). Transcripts were annotated using Kyoto Encyclopedia of Genes and Genomes (KEGG) analysis (*Kanehisa et al., 2008*) and Gene Ontology (GO). The GO annotation of genes was obtained using BLast2GO software. DEGs were enriched via KEGG and GO database. A Fisher exact test was used to find the vital enrichment pathway in the study by taking the significance of $p$-value < 0.05 and FDR < 0.05 as thresholds.

## Identification of candidate transcripts

The tBLASTn program was performed with available sequences of OBPs, CSPs, ORs IRs and SNMPs from other species as a "query" to identify candidate genes that encoded putative OBPs, CSPs, ORs, IRs and SNMPs in *A. granhmi*, respectively. All candidate OBPs, CSPs, ORs, IRs and SNMPs were manually checked by the BLASTp (http://blast.ncbi.nlm.nih.gov/Blast.cgi) search application. Soon after, the prediction

opening reading frame (ORF) of the candidate OBPs, CSPs, ORs, IRs and SNMPS genes was identified by the ORFfinder (https://www.ncbi.nlm.nih.gov/orffinder/). Conserved domain was predicted utilizing Batch CD-search (https://www.ncbi.nlm.nih.gov/Structure/bwrpsb/bwrpsb.cgi). The transmembrane domains (TMDs) of IRs, ORs and SNMPs were predicted by TMHMM server (v2.0) (http://www.cbs.dtu.dk/services/TMHMM/). The signal peptide of putative OBPs and CSPs were predicted using Signa1P (v5.0) (http://www.cbs.dtu.dk/services/SignalP/) server version with the default parameters.

## Quantitative real-time PCR analysis

To compare the differential expression of chemosensory genes between female and male antennal transcriptomes in *A. grahami*, the reads number of each olfactory-related gene was converted to FPKM (*Livak & Schmittgen, 2001*). Quantitative real-time PCR analysis (qRT-PCR) was performed to quantify the expression levels of olfactory-related genes in male and female antennae with rp49 as the reference gene (*Rodrigues et al., 2017*). Total RNA was extracted from 50 antennae obtained from females and males separately as described in the section of RNA isolation and quality assessment. The cDNA from antennae of both sexes was synthesized using the Goldenstar^{TM} RT6 cDNA Synthesis Mix. One µg of total RNA from samples was used in reverse transcription in a total volume of 20 µL reaction system to obtain the first-stand cDNA. The qRT-PCR was performed on an ABI 7500 using SYBR green dye (2×T5 Fast qPCR Mix) binding to double stranded DNA at the end of each elongation cycle. Primer sequences were designed by the Primer Premier 5.0 program (Table S1). QRT-PCR was conducted using previous method (*Jia et al., 2016*). In order to check reproducibility, qRT-PCR test for each sample was performed with three technical replicates and three biological replicates.

The Relative quantification analyses among samples were performed using comparative $2^{-\Delta\Delta Ct}$ method (*Schmittgen & Livak, 2008*).

## Tissue expression analysis

The expression of OBPs, ORs and non-olfactory genes of different organs were evaluated by qRT-PCR using the same procedure as the one for quantitative real-time PCR analysis. Female antennae (FA), leg (FL), wing (FW), head (FH) and male antennae (MA), leg (ML), wing (MW), head (MH) were collected from adult *A. grahami*. The Relative quantification analyses among samples were also performed using comparative $2^{-\Delta\Delta Ct}$ method.

## Phylogenetic analysis

Protein sequences of *Drosophila melanogaster*, *Calliphora stygia*, *Musca domestica* and *Lucilia cuprina* were obtained from Uniprot (*Apweiler et al., 2004*). The phylogenetic development of OBPs, ORs IRs, SNMPs and CSP trees were constructed by MEGA 5.0 with neighbor-joining method utilizing default setting and 1,000 bootstraps respectively (*Li et al., 2017*).

## RESULTS

### Overview of transcriptomes

An average of 6.46 GB raw data were generated from each sample by reading the male and the female's antenna samples. After removing adaptor sequences, low quality sequences, and N-containing sequences, a mean of 48.35 Mb clean reads were obtained for each sample (Table S2). The mean Q30 was about 89.0% for each sample. Clean reads from six samples, 74.83–78.58% were successfully mapped against the reference *A. grahami* genome (SRA: PRJNA513084). The percentage of unique mapping reads was from 49.82% to 51.92% in each sample. Pearson Correlation Coefficient between three biological replicates from the two groups (male and female) had high repeatability (i.e., all $R^2 \geq 0.9731$).

### Gene prediction and annotation

A total of 14,193 genes were annotated, including 11,327 known genes and 2,866 predicted novel genes. A total of 16,995 new transcripts were found, of which 9,460 belonged to newly alternative splicing subtypes of known protein coding genes, 2,951 belonged to transcripts of new protein coding genes, and the remaining 4,584 were long-noncoding RNA.

A total of 9,667 genes were enriched by GO annotation. GO divided genes into three categories, representing the molecular functions, cellular component, and biological processes, respectively. The first three categories that contain the largest number of genes are represented in Fig.1. The category includes binding (4,708), cell (3,617) and cellular process (3,549).

In total, 7,243 were functionally clustered into six KEGG categories including cellular processes, environmental and genetic information processing, and diseases and metabolism. Among the 44 sub-categories, "signal transduction" (1,677) and "global and overview maps" (1,397) were the most enriched (Fig. S1).

Moreover, 195 candidate olfactory related transcripts were found based on the results of annotation, including 36 OBPs (Table 1), 70 ORs (Table 2), seven IRs (Table S3), one CSP (Table S4), two SNMPs (Table S5) and nine non-olfactory genes (Fig. S2).

### Identification of olfactory genes of *A. grahami*

In total, 36 putative OBPs encoding sequences were found in the antennal transcriptome (Table 1) with five having no signal peptide. A total of 28 putative OBPs transcripts have intact open reading frames (ORF) with the lengths ranging from 100 to 300 aa. After analysis, out of the 28 genes 26 were selected with predicted domain belonging to pheromone/general odorant-binding protein (PhBP or PBP_GOBP) family. According to the number and sites of conserved cysteines, 21 *A. grahami* OBPs transcripts shared structural characteristics of OBPs (i.e., having typical six conserved cysteines) with other insects (*Jia et al., 2016*), including four "Minus-C" OBPs with C missing, and one "Plus-C" OBPs with more than six conserved cysteines and a proline (Fig. 2).

Seventy candidate ORs transcripts were found in the results of annotation (Table 2). Of these, 49 genes which contained 3–8 transmembrane domains (TMDs) and conservative domain (7tm_6 superfamily) were selected for further analysis. Seven putative IRs were identified in both male and female *A. grahami* antennal transcriptomes (Table S3) including

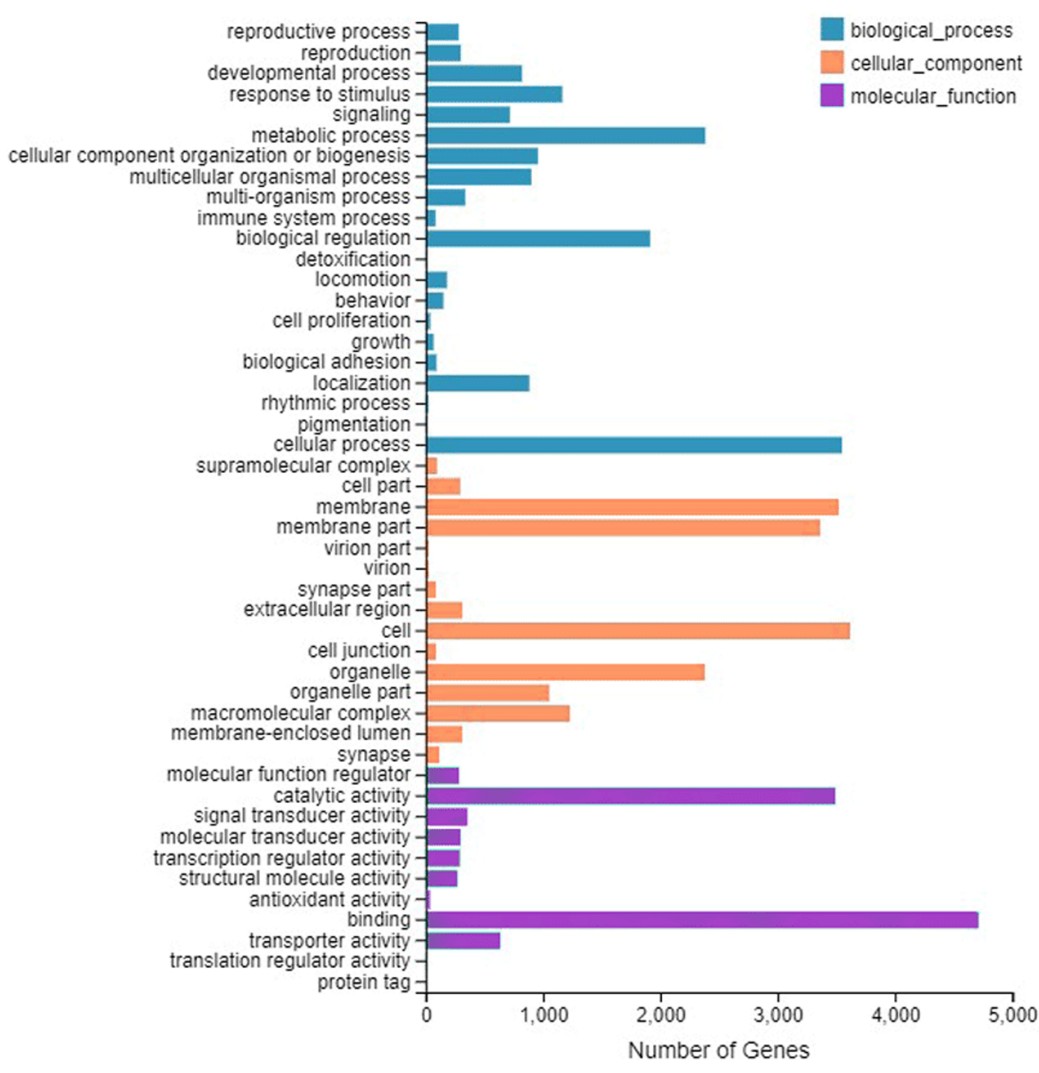

**Figure 1 Gene Ontology (GO) classification analysis of the *A. grahami* antennal different expression genes.** Unigenes are classified into three categories: biological process, cellular component, and molecular function. GO functions are shown on the *y*-axis. The *x*-axis shows the number of genes that have a particular GO function.

six IRs had 3–8 TMDs. One transcript encoding putative CSP (*A. grahami* OF01513) (Table S4) had a signal peptide and only two conserved cysteine residues. Two transcripts (*A. grahami* OF05479 and OF07379) which contained two TMDs with 450–600 aa were identified as SNMPs (Table S5).

## Gene expression differences between female and male

Based on the RNA-Seq by Expectation Maximization selection, 740 DEGs were chosen between female and male antennae. In total, 357 up-regulated and 383 down-regulated genes were found in male antennae. Among the DEG results, 21 differentially expressed olfactory genes, including four OBPs, eight ORs and nine other genes (which were enriched in the term of olfactory transduction function in KEGG) were found between the females and the males. The $\log_2^{(Aldrich\_M/Aldrich\_F)}$ ranged from 1.00 to 11.01 in the

**Table 1 Unigenes of candidate odorant binding proteins in *A. grahami*.**

| Gene name | Length (nt) | ORF (aa) | Status | Signal Peptide | E-value | Best Blastp hit | Domains | Superfamily | Aldrich F1 FPKM | Aldrich F2 FPKM | Aldrich F3 FPKM | Aldrich M1 FPKM | Aldrich M2 FPKM | Aldrich M3 FPKM |
|---|---|---|---|---|---|---|---|---|---|---|---|---|---|---|
| OF00694 | 5,063 | 171 | Complete | Y | 2E-108 | AID61304.1\| odorant binding protein (*Calliphora stygia*) | PhBP | cl11600 | 6,793.97 | 6,491.5 | 6,538.27 | 7,557.2 | 7,495.94 | 8,220.24 |
| OF01908 | 1,908 | 148 | Complete | Y | 1E-96 | AID61299.1\| odorant binding protein (*Calliphora stygia*) | PhBP | cl11600 | 4.55 | 1.15 | 2.66 | 0.84 | 1.79 | 3.18 |
| OF03173 | 336 | 107 | N | N | 3E-12 | XP_005189169.1\| PREDICTED: general odorant-binding protein 57c-like (*Musca domestica*) | PhBP | cl11600 | 0 | 0 | 0 | 0 | 0.41 | 0 |
| OF03727 | 528 | 150 | Complete | Y | 3E-102 | AID61318.1\| odorant binding protein (*Calliphora stygia*) | PBP_GOBP | cl11600 | 449.22 | 330.2 | 451.67 | 622.38 | 425.39 | 481.57 |
| OF03728 | 542 | 150 | Complete | Y | 1E-78 | AID61320.1\| odorant binding protein (*Calliphora stygia*) | PBP_GOBP | cl11600 | 7.63 | 5.37 | 3.25 | 17.5 | 5.01 | 10.85 |
| OF03737 | 572 | 163 | Complete | Y | 3E-93 | AID61305.1\| odorant binding protein (*Calliphora stygia*) | PBP_GOBP | cl11600 | 109.56 | 102.35 | 176.49 | 225.97 | 64.39 | 211.97 |
| OF03989 | 5,496 | 144 | N | N | 7E-86 | KNC25071.1\| Pheromone-binding protein-related protein 3 (*Lucilia cuprina*) | PhBP | cl11600 | 5,731.45 | 5,449.59 | 7,028.92 | 5,625.04 | 6,036.15 | 5,382.4 |
| OF03991 | 1,523 | 152 | N | Y | 9E-105 | AID61303.1\| odorant binding protein (*Calliphora stygia*) | PBP_GOBP | cl11600 | 2,996.22 | 2,605.5 | 3,211.17 | 3,989.71 | 3,439.09 | 3,421.55 |
| OF04509 | 743 | 200 | N | N | 4E-84 | AID61317.1\| odorant binding protein, partial (*Calliphora stygia*) | – | – | 1.48 | 2.48 | 1.9 | 2.75 | 3.12 | 2.62 |

(Continued)

| Gene name | Length (nt) | ORF (aa) | Status | Signal Peptide | E-value | Best Blastp hit | Domains | Superfamily | Aldrich F1 FPKM | Aldrich F2 FPKM | Aldrich F3 FPKM | Aldrich M1 FPKM | Aldrich M2 FPKM | Aldrich M3 FPKM |
|---|---|---|---|---|---|---|---|---|---|---|---|---|---|---|
| OF06318 | 708 | 148 | Complete | Y | 4E−91 | XP_023303590.1\| general odorant-binding protein 28a-like (*Lucilia cuprina*) | PhBP | cl11600 | 3.25 | 0.68 | 3.97 | 2.3 | 1.05 | 2.48 |
| OF06321 | 1,883 | 148 | Complete | Y | 6E−97 | AID61299.1\| odorant binding protein (*Calliphora stygia*) | PhBP | cl11600 | 2.15 | 1.96 | 0.83 | 0.75 | 0 | 0.38 |
| OF06322 | 607 | 148 | Complete | Y | 1E−71 | AID61299.1\| odorant binding protein (*Calliphora stygia*) | PBP_GOBP | cl11600 | 16.29 | 18.22 | 14.52 | 15.05 | 16.09 | 19.27 |
| OF07356 | 538 | 135 | Complete | Y | 1E−83 | AID61306.1\| odorant binding protein, partial (*Calliphora stygia*) | PBP_GOBP | cl11600 | 0.2 | 0.21 | 0 | 0.2 | 0.22 | 0.21 |
| OF08065 | 741 | 143 | Complete | Y | 1E−76 | AID61298.1\| odorant binding protein (*Calliphora stygia*) | PBP_GOBP | cl11600 | 11,111.59 | 10,312.81 | 12,733.79 | 12,913.21 | 10,822.75 | 12,523.44 |
| OF08070 | 865 | 111 | N | Y | 7E−62 | XP_023299671.1\| general odorant-binding protein 19d-like (*Lucilia cuprina*) | PBP_GOBP superfamily | – | 3.67 | 2.67 | 2.48 | 4.19 | 2.05 | 1.71 |
| OF08228 | 1,720 | 141 | Complete | Y | 2E−93 | AID61300.1\| odorant binding protein (*Calliphora stygia*) | PBP_GOBP | cl11600 | 6.71 | 9.04 | 6.23 | 82.13 | 10.14 | 14.36 |
| OF08714 | 540 | 153 | Complete | Y | 5E−99 | AID61319.1\| odorant binding protein, partial (*Calliphora stygia*) | PBP_GOBP | cl11600 | 284.68 | 298.87 | 337.26 | 231.69 | 180.27 | 219.17 |
| OF08934 | 592 | 147 | Complete | Y | 2E−63 | KNC27005.1\| Pheromone-binding protein-related protein 5 (*Lucilia cuprina*) | PhBP | cl11600 | 0 | 0 | 0 | 0.18 | 0 | 0 |

| Gene name | Length (nt) | ORF (aa) | Status | Signal Peptide | E-value | Best Blastp hit | Domains | Superfamily | Aldrich F1 FPKM | Aldrich F2 FPKM | Aldrich F3 FPKM | Aldrich M1 FPKM | Aldrich M2 FPKM | Aldrich M3 FPKM |
|---|---|---|---|---|---|---|---|---|---|---|---|---|---|---|
| OF11577 | 3,511 | 189 | Complete | Y | 2E−129 | AID61316.1\| odorant binding protein (*Calliphora stygia*) | – | – | 1.73 | 1.16 | 0.28 | 1.86 | 2.89 | 1.75 |
| OF12159 | 476 | 135 | Complete | Y | 1E−88 | AID61309.1\| odorant binding protein (*Calliphora stygia*) | PBP_GOBP | cl11600 | 57.97 | 29.38 | 33.06 | 206.02 | 63.39 | 37.7 |
| OF12160 | 478 | 118 | Complete | Y | 9E−62 | AID61309.1\| odorant binding protein (*Calliphora stygia*) | PBP_GOBP | cl11600 | 2,224.07 | 1,490.12 | 1,146.79 | 3,112.62 | 1,741.97 | 1,159.57 |
| OF12161 | 540 | 136 | Complete | Y | 4E−82 | XP_023297832.1\| general odorant-binding protein 56a-like (*Lucilia cuprina*) | PBP_GOBP | cl11600 | 0 | 0.21 | 0 | 0 | 0.22 | 0 |
| OF12162 | 484 | 136 | Complete | Y | 3E−79 | XP_023297815.1\| general odorant-binding protein 56a-like (*Lucilia cuprina*) | PBP_GOBP | cl11600 | 0 | 0 | 0 | 0.23 | 0 | 0 |
| OF12163 | 484 | 136 | Complete | Y | 4E−82 | XP_023297832.1\| general odorant-binding protein 56a-like (*Lucilia cuprina*) | PBP_GOBP | cl11600 | 0 | 0 | 0 | 0.23 | 0 | 0 |
| OF12164 | 523 | 138 | Complete | Y | 8E−84 | XP_023297815.1\| general odorant-binding protein 56a-like (*Lucilia cuprina*) | PBP_GOBP | cl11600 | 0 | 0 | 0 | 0.21 | 0 | 0 |
| OF12165 | 518 | 136 | Complete | Y | 8E−79 | AID61314.1\| odorant binding protein, partial (*Calliphora stygia*) | PBP_GOBP | cl11600 | 17.14 | 4.38 | 4.86 | 16.54 | 6.3 | 7.2 |
| OF12166 | 507 | 134 | Complete | Y | 6E−72 | AID61314.1\| odorant binding protein, partial (*Calliphora stygia*) | PBP_GOBP | cl11600 | 6.88 | 0.6 | 1.46 | 1.43 | 0.31 | 5.88 |

(Continued)

| Gene name | Length (nt) | ORF (aa) | Status | Signal Peptide | E-value | Best Blastp hit | Domains | Superfamily | Aldrich F1 FPKM | Aldrich F2 FPKM | Aldrich F3 FPKM | Aldrich M1 FPKM | Aldrich M2 FPKM | Aldrich M3 FPKM |
|---|---|---|---|---|---|---|---|---|---|---|---|---|---|---|
| OF12594 | 430 | 116 | Complete | N | 3E–80 | AID61296.1| odorant binding protein (Calliphora stygia) | PBP_GOBP | cl11600 | 7,264.31 | 7,580.12 | 9,447.39 | 6,675.22 | 5,594.38 | 6,033.82 |
| OF06267 | 691 | 207 | N | Y | 5E–106 | XP_023306331.1| putative odorant-binding protein A5 (Lucilia cuprina) | PEBP_euk | cl00227 | 4,579.82 | 3,726.1 | 4,497.4 | 7,292.1 | 7,843.41 | 6,500.85 |
| OF08890 | 530 | 146 | N | Y | 8E–102 | AID61325.1| chemosensory protein (Calliphora stygia) | OS-D | cl04042 | 53,230.86 | 42,911.58 | 55,432.39 | 46,095.65 | 51,606.36 | 50,779.65 |
| OF11575 | 968 | 295 | N | N | 2E–81 | XP_023301477.1| general odorant-binding protein 71 (Lucilia cuprina) | – | – | 201.94 | 299.18 | 244.47 | 286.73 | 303.85 | 291.82 |
| OF09681 | 27,357 | 139 | Complete | Y | 3E–50 | AID61301.1| odorant binding protein (Calliphora stygia) | PBP_GOBP | cl11600 | 2.91 | 2.3 | 3.79 | 11.58 | 15.05 | 14.1 |
| OF08066 | 709 | 143 | Complete | Y | 4E–64 | AID61308.1| odorant binding protein (Calliphora stygia) | PBP_GOBP | cl11600 | 23.69 | 27.51 | 50.12 | 46.85 | 52.39 | 66.91 |
| OF08068 | 7,857 | 143 | Complete | Y | 1E–87 | AID61308.1| odorant binding protein (Calliphora stygia) | PBP_GOBP | cl11600 | 0 | 0 | 0 | 0 | 0.01 | 0.01 |
| OF08069 | 4,971 | 143 | Complete | Y | 4E–74 | KNC26975.1| Pheromone-binding protein-related protein 2 (Lucilia cuprina) | PBP_GOBP | cl11600 | 3,791.25 | 3,317.74 | 3,754.29 | 4,382.82 | 3,998.67 | 4,826.81 |
| OF10114 | 705 | 177 | Complete | Y | 2E–74 | KNC33289.1| hypothetical protein FF38_06045, partial (Lucilia cuprina) | – | – | 0 | 0.3 | 0 | 0 | 0 | 0.3 |

**Note:**
ORF, opening read frame; FPKM, fragments per kilobase of transcript per million mapped reads.

**Table 2 Unigenes of odorant receptors proteins in A. grahami.**

| Gene name | Length (nt) | Amino acid length (aa) | TMDs | E-value | BLASTx best hit | Domains | Superfamily | Aldrich F1 FPKM | Aldrich F2 FPKM | Aldrich F3 FPKM | Aldrich M1 FPKM | Aldrich M2 FPKM | Aldrich M3 FPKM |
|---|---|---|---|---|---|---|---|---|---|---|---|---|---|
| OF00033 | 1,355 | 370 | 6 | 0 | AID61240.1|odorant receptor, partial (Calliphora stygia) | 7tm_6 superfamily | | 6.91 | 4.42 | 7.13 | 6.71 | 9.53 | 8.73 |
| OF03291 | 22,874 | 662 | 8 | 3.00E−170 | XP_023303346.1|odorant receptor 7a-like (Lucilia cuprina) | 7tm_6 | cl20237 | 2.03 | 2.07 | 2.12 | 1.09 | 1.43 | 1.6 |
| OF09092 | 6,367 | 750 | 12 | 0 | KNC26770.1|putative odorant receptor 7a (Lucilia cuprina) | 7tm_6 | cl20237 | 80.79 | 60.23 | 73.3 | 44.91 | 33.22 | 31.48 |
| OF09093 | 10,862 | 430 | 4 | 0 | AID61204.1|odorant receptor, partial (Calliphora stygia) | 7tm_6 | cl20237 | 0.2 | 0.23 | 0.13 | 0.15 | 0.17 | 0.1 |
| OF09094 | 1,419 | 407 | 6 | 0 | XP_023291420.1|odorant receptor 42b-like (Lucilia cuprina) | 7tm_6 | cl20237 | 0.06 | 0.13 | 0.13 | 0.07 | 0.35 | 0 |
| OF09095 | 3,759 | 433 | 7 | 0 | AID61230.1|odorant receptor (Calliphora stygia) | 7tm_6 | cl20237 | 93.55 | 66.21 | 86.56 | 117.23 | 135.59 | 75.1 |
| OF02163 | 5,901 | 431 | 6 | 0 | AID61219.1|odorant receptor, partial (Calliphora stygia) | 7tm_6 superfamily | – | 15.19 | 11.05 | 11.37 | 25.03 | 28.1 | 21.15 |
| OF02164 | 5,239 | 271 | 4 | 6.00E−159 | XP_023305776.1|odorant receptor 63a-like (Lucilia cuprina) | 7tm_6 superfamily | – | 21.83 | 21.73 | 21.82 | 32.99 | 38.56 | 33.01 |
| OF12341 | 9,653 | 432 | 7 | 0 | XP_023306197.1|odorant receptor 63a-like (Lucilia cuprina) | 7tm_6 superfamily | – | 0 | 0.02 | 0 | 0 | 0 | 0 |
| OF12296 | 5,257 | 489 | 6 | 0 | AID61256.1|gustatory receptor (Calliphora stygia) | 7tm_7 | cl19976 | 17.78 | 11.27 | 17.35 | 17.55 | 26.16 | 16.81 |
| OF12237 | 1,011 | 314 | 5 | 0 | AID61205.1|odorant receptor, partial (Calliphora stygia) | 7tm_6 superfamily | – | 21.96 | 22.55 | 22.87 | 19.26 | 32.51 | 28.36 |
| OF05116 | 9,171 | 441 | 4 | 0 | KNC34511.1|putative odorant receptor 13a (Lucilia cuprina) | 7tm_6 superfamily | – | 0.02 | 0.01 | 0.02 | 0 | 0 | 0.01 |
| OF08895 | 1,732 | 434 | 2 | 0 | XP_023299035.1|odorant receptor 13a (Lucilia cuprina) | 7tm_6 superfamily | – | 6.41 | 4.22 | 5.83 | 14.64 | 8.2 | 6.38 |
| OF04344 | 21,164 | 425 | 6 | 0 | AID61222.1|odorant receptor (Calliphora stygia) | 7tm_6 superfamily | – | 17.3 | 14.93 | 11.93 | 18.02 | 17.93 | 12.12 |
| OF03613 | 1,357 | 364 | 4 | 0 | XP_023294606.1|odorant receptor 67d-like (Lucilia cuprina) | 7tm_6 superfamily | – | 41.35 | 34.72 | 41.71 | 51.6 | 67.66 | 59.32 |

(Continued)

| Gene name | Length (nt) | Amino acid length (aa) | TMDs | E-value | BLASTx best hit | Domains | Superfamily | Aldrich F1 FPKM | Aldrich F2 FPKM | Aldrich F3 FPKM | Aldrich M1 FPKM | Aldrich M2 FPKM | Aldrich M3 FPKM |
|---|---|---|---|---|---|---|---|---|---|---|---|---|---|
| OF03614 | 1,694 | 389 | 4 | 0 | XP_023294606.1|odorant receptor 67d-like (*Lucilia cuprina*) | 7tm_6 superfamily | – | 23.77 | 20.69 | 24.16 | 34.77 | 42.81 | 29.3 |
| OF05346 | 1,378 | 389 | 4 | 0 | AID61234.1|odorant receptor, partial (*Calliphora stygia*) | 7tm_6 superfamily | – | 6.29 | 5.63 | 5.55 | 7.44 | 8.26 | 5.33 |
| OF05347 | 1,382 | 367 | 6 | 0 | AID61245.1|odorant receptor (*Calliphora stygia*) | 7tm_6 superfamily | – | 16.43 | 14.6 | 19.38 | 88.9 | 102.27 | 70.03 |
| OF05348 | 1,381 | 388 | 6 | 0 | AID61234.1|odorant receptor, partial (*Calliphora stygia*) | 7tm_6 superfamily | – | 117.78 | 96.55 | 121.5 | 212.3 | 218.08 | 182.72 |
| OF05349 | 1,442 | 387 | 6 | 0 | XP_023305356.1|odorant receptor 67d-like (*Lucilia cuprina*) | 7tm_6 superfamily | – | 10.74 | 9.71 | 7.41 | 29.22 | 29.36 | 16.51 |
| OF08236 | 6,012 | 400 | 6 | 0 | XP_023302956.1|odorant receptor 67c (*Lucilia cuprina*) | 7tm_6 superfamily | – | 0.1 | 0.06 | 0.04 | 0 | 0 | 0.03 |
| OF04333 | 6,837 | 397 | 8 | 0 | XP_023300158.1|putative odorant receptor 92a (*Lucilia cuprina*) | 7tm_6 superfamily | – | 80.87 | 58.32 | 77.81 | 126.74 | 142.08 | 114.04 |
| OF05773 | 2,812 | 440 | 4 | 0 | AID61252.1|gustatory receptor (*Calliphora stygia*) | 7tm_7 | cl19976 | 57.63 | 47.34 | 60.21 | 65.2 | 72.07 | 63.46 |
| OF07668 | 26,554 | 478 | 7 | 0 | AID61201.1|odorant receptor, partial (*Calliphora stygia*) | 7tm_6 superfamily | – | 653.15 | 710.04 | 643.17 | 604.21 | 753.33 | 653.29 |
| OF09503 | 3,710 | 451 | 6 | 0 | XP_023301661.1|odorant receptor 83a (*Lucilia cuprina*) | 7tm_6 superfamily | – | 0 | 0.02 | 0 | 0 | 0 | 0 |
| OF09504 | 36,020 | 1,377 | 18 | 0 | KNC30410.1|putative odorant receptor 83a (*Lucilia cuprina*) | 7tm_6 superfamily | – | 0.05 | 0.05 | 0.03 | 0.03 | 0.04 | 0.05 |
| OF00360 | 3,606 | 362 | 4 | 0 | AID61207.1|odorant receptor (*Calliphora stygia*) | 7tm_6 superfamily | – | 4.05 | 3.66 | 5.59 | 7.23 | 5.61 | 4.88 |
| OF00361 | 1,460 | 359 | 4 | 0 | XP_023295767.1|odorant receptor 30a-like (*Lucilia cuprina*) | 7tm_6 | cl20237 | 18.1 | 14.39 | 17.19 | 39.55 | 35.98 | 21.95 |
| OF03270 | 6,328 | 431 | 4 | 0 | KNC22013.1|putative odorant receptor 85b (*Lucilia cuprina*) | 7tm_6 | cl20237 | 0.64 | 0.46 | 0.49 | 0.1 | 0.03 | 0.09 |

| Gene name | Length (nt) | Amino acid length (aa) | TMDs | E-value | BLASTx best hit | Domains | Superfamily | Aldrich F1 FPKM | Aldrich F2 FPKM | Aldrich F3 FPKM | Aldrich M1 FPKM | Aldrich M2 FPKM | Aldrich M3 FPKM |
|---|---|---|---|---|---|---|---|---|---|---|---|---|---|
| OF03271 | 1,412 | 406 | 3 | 0 | XP_023301736.1\|odorant receptor 85b-like (*Lucilia cuprina*) | 7tm_6 | cl20237 | 0.19 | 0.67 | 0.85 | 0 | 0 | 0.07 |
| OF03272 | 8,161 | 430 | 4 | 0 | KNC22013.1\|putative odorant receptor 85b (*Lucilia cuprina*) | 7tm_6 | cl20237 | 0 | 0 | 0.02 | 0.01 | 0 | 0 |
| OF10424 | 2,968 | 397 | 6 | 0 | XP_023303249.1\|putative odorant receptor 85d (*Lucilia cuprina*) | 7tm_6 | cl20237 | 0.09 | 0 | 0.12 | 0.03 | 0.06 | 0.09 |
| OF04655 | 1,425 | 409 | 4 | 0 | AID61214.1\|odorant receptor, partial (*Calliphora stygia*) | 7tm_6 superfamily | – | 63.51 | 54.65 | 62.79 | 152.99 | 153.77 | 119.31 |
| OF01050 | 1,401 | 401 | 6 | 0 | AID61239.1\|odorant receptor (*Calliphora stygia*) | 7tm_6 superfamily | – | 201.87 | 159.01 | 201.63 | 169.82 | 156.98 | 116.52 |
| OF07350 | 3,826 | 400 | 6 | 0 | AID61209.1\|odorant receptor (*Calliphora stygia*) | 7tm_6 | cl20237 | 553.8 | 565.58 | 537.74 | 370.78 | 541.56 | 439.42 |
| OF10697 | 6,210 | 382 | 6 | 0 | TMW44128.1\|hypothetical protein DOY81_010793 (*Sarcophaga bullata*) | 7tm_6 superfamily | – | 16.45 | 16.59 | 16.2 | 8.39 | 13.07 | 11.52 |
| OF11806 | 3,426 | 371 | 6 | 0 | AID61215.1\|odorant receptor (*Calliphora stygia*) | 7tm_6 superfamily | – | 11.42 | 16.36 | 14.73 | 16.38 | 19.36 | 22.52 |
| OF08715 | 18,470 | 388 | 6 | 0 | XP_023292635.1\|odorant receptor 45a-like (*Lucilia cuprina*) | 7tm_6 superfamily | – | 0.02 | 0.03 | 0.04 | 0.03 | 0.03 | 0.03 |
| OF10138 | 1,444 | 400 | 7 | 4.00E−92 | KNC21829.1\|putative odorant receptor 45a (*Lucilia cuprina*) | 7tm_6 superfamily | – | 0 | 0.07 | 0 | 0 | 0 | 0.06 |
| OF11636 | 3,805 | 418 | 7 | 0 | AID61244.1\|odorant receptor (*Calliphora stygia*) | 7tm_6 superfamily | – | 6.47 | 5.54 | 5.81 | 1.01 | 1.67 | 1.42 |
| OF11637 | 4,863 | 434 | 7 | 0 | AID61243.1\|odorant receptor (*Calliphora stygia*) | 7tm_6 superfamily | – | 42.02 | 38.13 | 43.94 | 58.6 | 57.21 | 53.83 |
| OF11638 | 3,925 | 395 | 4 | 0 | AID61236.1\|odorant receptor, partial (*Calliphora stygia*) | 7tm_6 superfamily | – | 10.74 | 12.37 | 11.41 | 13.63 | 13.22 | 15.09 |
| OF03087 | 44,440 | 215 | 2 | 1.00E−90 | AID61265.1\|gustatory receptor (*Calliphora stygia*) | 7tm_7 superfamily | – | 0.02 | 0.02 | 0.04 | 0.02 | 0.02 | 0.02 |
| OF04907 | 5,976 | 382 | 6 | 0 | AID61224.1\|odorant receptor (*Calliphora stygia*) | GT1 superfamily | – | 35.9 | 25.1 | 31.57 | 57.74 | 75.76 | 56.73 |
| OF00577 | 2,407 | 461 | 3 | 0 | XP_023290912.1\|putative odorant receptor 85e (*Lucilia cuprina*) | 7tm_6 superfamily | – | 0 | 0 | 0 | 0 | 0.04 | 0 |

(Continued)

| Gene name | Length (nt) | Amino acid length (aa) | TMDs | E-value | BLASTx best hit | Domains | Superfamily | Aldrich F1 FPKM | Aldrich F2 FPKM | Aldrich F3 FPKM | Aldrich M1 FPKM | Aldrich M2 FPKM | Aldrich M3 FPKM |
|---|---|---|---|---|---|---|---|---|---|---|---|---|---|
| OF09932 | 5,332 | 389 | 6 | 0 | AID61217.1\|odorant receptor (Calliphora stygia) | 7tm_6 | cl20237 | 5.09 | 4.15 | 4.68 | 5.89 | 8.35 | 5.73 |
| OF09933 | 7,816 | 393 | 6 | 0 | AID61218.1\|odorant receptor (Calliphora stygia) | 7tm_6 | cl20237 | 9.4 | 7.99 | 10.61 | 6.87 | 7.24 | 6.25 |
| OF11666 | 1,703 | 392 | 6 | 0 | AID61212.1\|odorant receptor, partial (Calliphora stygia) | 7tm_6 superfamily | – | 6.13 | 8.42 | 7.7 | 11.59 | 8.84 | 6.57 |
| OF11667 | 745 | 245 | 3 | 1.00E−103 | XP_023296402.1\|odorant receptor 46a-like (Lucilia cuprina) | 7tm_6 superfamily | – | 0 | 0.14 | 0.14 | 0 | 0.14 | 0 |
| OF00896 | 4,041 | 347 | 6 | 0 | AID61211.1\|odorant receptor (Calliphora stygia) | 7tm_6 superfamily | – | 108.76 | 103.46 | 114.6 | 57.1 | 75.02 | 68.08 |
| OF00900 | 7,334 | 375 | 7 | 0 | AID61210.1\|odorant receptor (Calliphora stygia) | 7tm_6 | cl20237 | 37.52 | 22.67 | 30.74 | 51.14 | 54.37 | 41.49 |
| OF00084 | 112,602 | 1,206 | 1 | 0 | XP_023300977.1\|uncharacterized protein LOC111683162 (Lucilia cuprina) | Ig superfamily | – | 0.01 | 0.02 | 0.02 | 0.01 | 0.02 | 0.01 |
| OF00731 | 35,001 | 311 | 4 | 0 | XP_023301990.1\|transmembrane protein 47 isoform X1 (Lucilia cuprina) | – | – | 0.04 | 0.02 | 0.01 | 0.01 | 0.02 | 0.02 |
| OF00834 | 2,837 | 705 | 0 | 0 | XP_023301258.1\|putative uncharacterized protein DDB_G0292292 (Lucilia cuprina) | alpha-crystallin-Hsps_p23-like superfamily | – | 4.6 | 5.12 | 3.6 | 2.59 | 3.94 | 4.57 |
| OF02467 | 73,065 | 564 | 0 | 0 | KNC22574.1\|hypothetical protein FF38_00190, partial (Lucilia cuprina) | Ig_3 | cl11960 | 0.01 | 0.02 | 0.02 | 0.01 | 0.01 | 0.01 |
| OF02745 | 2,361 | 393 | 4 | 0 | AID61202.1\|odorant receptor (Calliphora stygia) | 7tm_6 | cl20237 | 16.12 | 12.4 | 14.12 | 14.48 | 14.77 | 14.68 |
| OF02968 | 12,261 | 223 | 1 | 1.00E−146 | XP_023309200.1\|uncharacterized protein LOC111690855 (Lucilia cuprina) | – | – | 277.64 | 277.72 | 344.84 | 332.6 | 383.69 | 376.79 |
| OF04274 | 9,170 | 1,302 | 0 | 0 | XP_023302813.1\|uncharacterized protein LOC111684831 (Lucilia cuprina) | PTZ00280 superfamily | – | 0.06 | 0.05 | 0.03 | 0.02 | 0.01 | 0.07 |

| Gene name | Length (nt) | Amino acid length (aa) | TMDs | E-value | BLASTx best hit | Domains | Superfamily | Aldrich F1 FPKM | Aldrich F2 FPKM | Aldrich F3 FPKM | Aldrich M1 FPKM | Aldrich M2 FPKM | Aldrich M3 FPKM |
|---|---|---|---|---|---|---|---|---|---|---|---|---|---|
| OF04296 | 7,115 | 218 | 0 | 7.00E−136 | XP_023305855.1| uncharacterized protein LOC111687629 isoform X5 (*Lucilia cuprina*) | – | – | 0.25 | 0.54 | 0.24 | 0.27 | 0.2 | 0.34 |
| OF05495 | 140,370 | 767 | 11 | 0 | KNC33152.1|hypothetical protein FF38_03670 (*Lucilia cuprina*) | Na_H_Exchanger | cl01133 | 0.01 | 0.01 | 0.01 | 0.01 | 0.01 | 0.01 |
| OF06445 | 2,695 | 107 | 0 | 6.00E−32 | KNC21873.1|hypothetical protein FF38_00598, partial (*Lucilia cuprina*) | – | – | 1.4 | 2.15 | 1.17 | 0.96 | 1.37 | 1.22 |
| OF06787 | 54,253 | 401 | 8 | 0 | XP_023292082.1|odorant receptor 22c (*Lucilia cuprina*) | 7tm_6 superfamily | – | 0.01 | 0.02 | 0.01 | 0.01 | 0.01 | 0.01 |
| OF07123 | 15,299 | 421 | 7 | 0 | AID61247.1|odorant receptor, partial (*Calliphora stygia*) | 7tm_6 superfamily | – | 0.97 | 1.24 | 0.89 | 0.94 | 1.23 | 1.66 |
| OF07686 | 66,956 | 648 | 0 | 0 | XP_023309492.1|adenylate kinase isoenzyme 5 (*Lucilia cuprina*) | ADK | cl17190 | 0.08 | 0.05 | 0.06 | 0.05 | 0.12 | 0.08 |
| OF10581 | 46,004 | 1,540 | 0 | 0 | KNC30311.1|hypothetical protein FF38_03958 (*Lucilia cuprina*) | Chorein_N | cl14987 | 0.04 | 0.05 | 0.01 | 0.02 | 0.03 | 0.03 |
| OF11048 | 5,109 | 147 | 0 | 4.00E−43 | XP_023301321.1|putative uncharacterized protein DDB_G0271606, partial (*Lucilia cuprina*) | – | – | 0 | 0 | 0 | 0 | 0 | 0.02 |
| OF11055 | 21,378 | 246 | 0 | 3.00E−147 | XP_023298603.1|leucine-rich repeat-containing protein 20 isoform X1 (*Lucilia cuprina*) | PLN00113 superfamily | – | 6.67 | 9.26 | 7.86 | 23.57 | 11.89 | 8.27 |
| OF11179 | 13,667 | 314 | 0 | 7.00E−164 | XP_023301587.1|ADP-ribosylation factor 1 (*Lucilia cuprina*) | P-loop_NTPase superfamily | cl02475 | 0.87 | 0.78 | 0.8 | 0.97 | 0.74 | 0.82 |
| OF11420 | 147,608 | 741 | 0 | 0 | KNC24737.1|LIM and SH3 domain protein Lasp (*Lucilia cuprina*) | LIM_LASP | – | 2.3 | 1.99 | 2.16 | 2.15 | 2.62 | 2.37 |
| OF11531 | 11,052 | 180 | 2 | 2.00E−112 | XP_023300793.1| uncharacterized protein LOC111683003 (*Lucilia cuprina*) | – | – | 2.12 | 2.25 | 2.34 | 2.4 | 2.63 | 2.33 |

**Note:**
TMDs, transmembrane domains; FPKM, fragments per kilobase of transcript per million mapped reads.

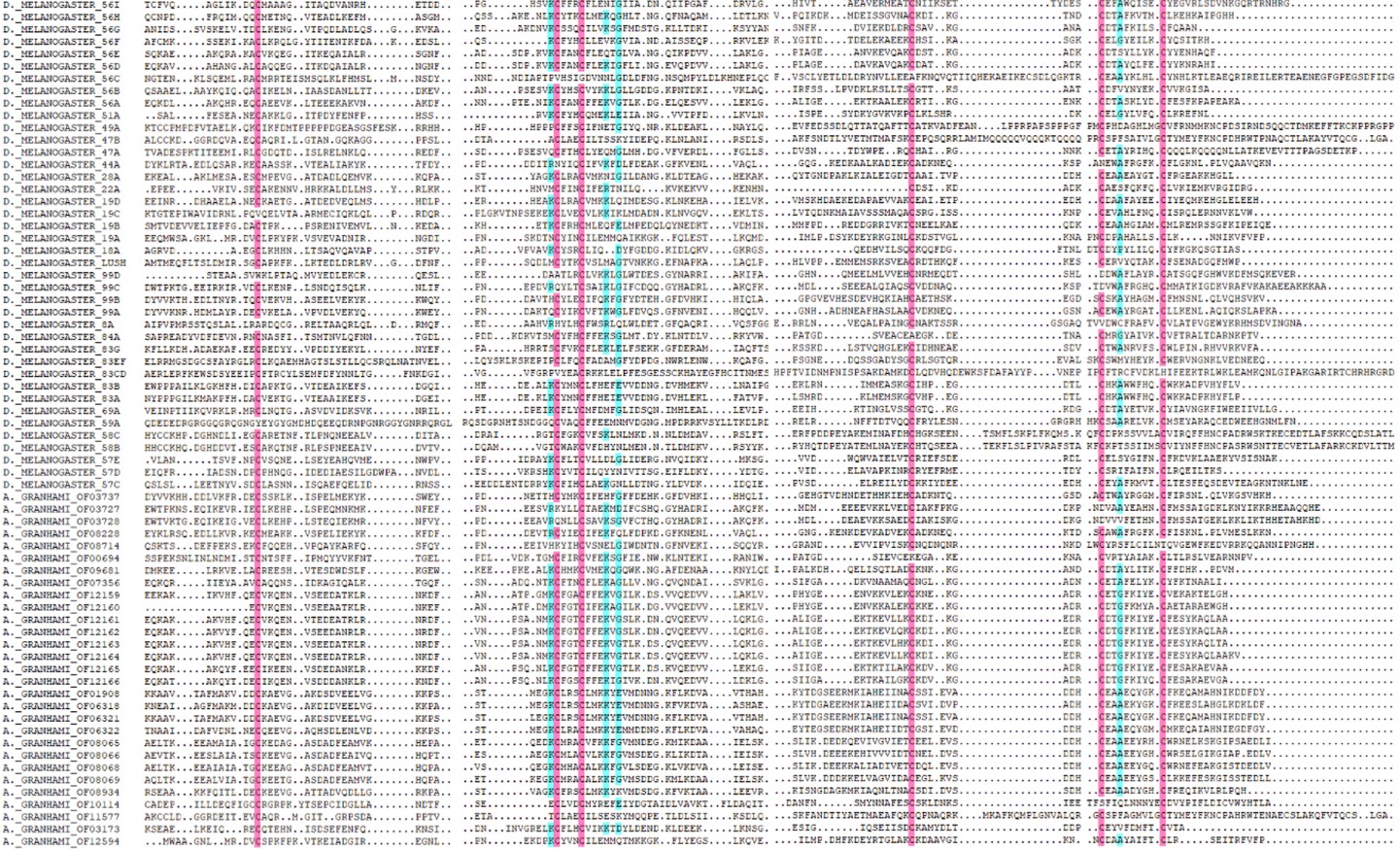

**Figure 2 Amino acid alignment of OBPs in *A. grahami* and OBPs sequences from *D. melanogaster*.** Amino acid sequences of *A. grahami* and *D. melanogaster* OBPs are aligned by DNAMAN. Blue and pink boxes show conserved cysteine.

up-regulated genes. Meanwhile, in the down-regulated genes, the $\log_2^{(\text{Aldrich\_M/Aldrich\_F})}$ ranged from −1.00 to −10.64.

## Sex-specific expression

Based on the transcriptome results, 76 genes exhibited sex-specific expression, among which 37 genes only found in male antennae and 39 genes only present in female antennae. In addition, among the 195 candidate olfactory genes, 10 genes were male-specific and seven were female-specific genes. These include five OBPs and three ORs male-specific and two ORs female-specific.

For qRT-PCR verification, we selected two male-specific and one female-specific gene with relatively high expression based on the FPKM. These genes were *A. grahami* OF03173, OF08934 and BGI_novel_G000488 in males, OF12341 in females, respectively. A total of 10 DEGs were selected for qRT-PCR verification (Figs. 3–5).

The qRT-PCR results supported the data obtained by transcriptome sequencing. The expression levels of three OBPs genes (*A. grahami* OF08228, OF03173 and OF08934) in male antennae were significantly upregulated in qRT-PCR results and consistent with those obtained by RNA-seq (Fig. 3), suggesting that these genes were gender specific. In addition,

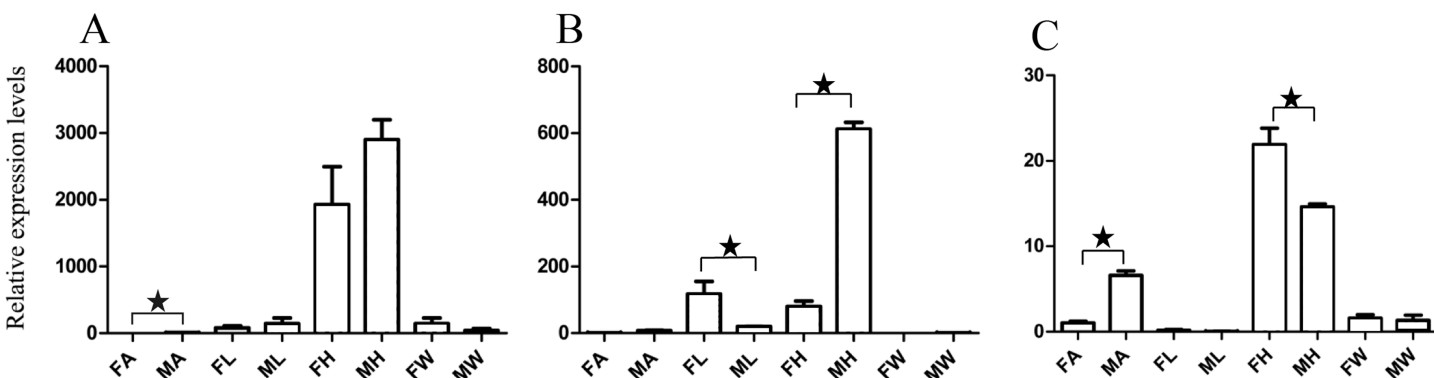

**Figure 3 Relative expression levels of A. grahami OBPs genes.** (A) *A. grahami* OF08228. (B) *A. grahami* OF03173. (C) *A. grahami* OF08934. FA, female antennae; MA, male antennae; FL, female leg; ML, male leg; FH, female head; MH, male head; FW, female wing; MW, male wing. The error bar represents standard error and the different "*" above each bar indicate significant differences in transcript abundances (*$p < 0.05$).

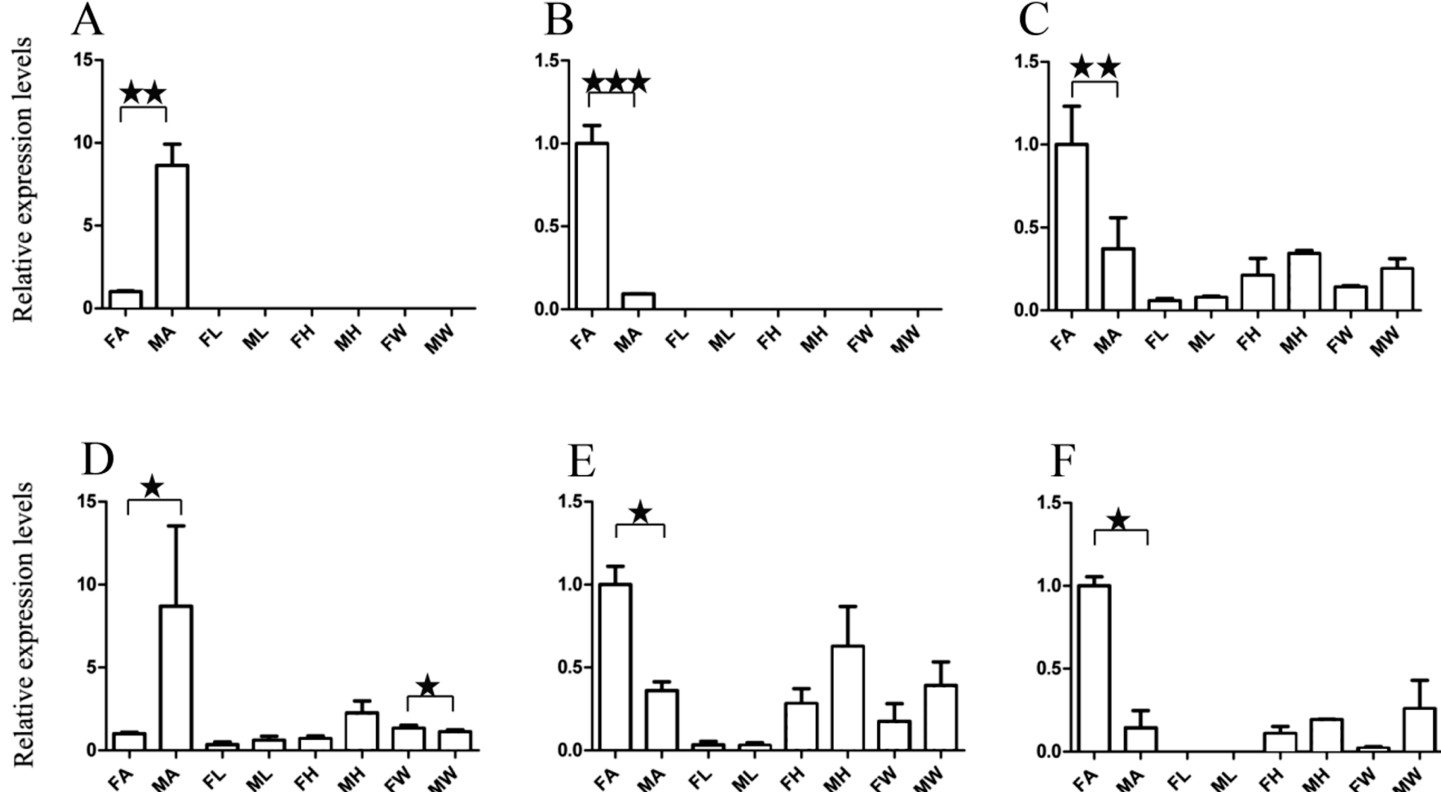

**Figure 4 Relative expression levels of A. grahami ORs genes.** (A) *A. grahami* OF05347. (B) *A. grahami* OF11636. (C) *A. grahami* OF03270. (D) *A. grahami* BGI_novel_G000488. (E) *A. grahami* OF12341. (F) *A. grahami* OF03271. FA, female antennae; MA, male antennae; FL, female leg; ML, male leg; FH, female head; MH, male head; FW, female wing; MW, male wing. The error bar represents standard error and the different "*" above each bar indicate significant differences in transcript abundances (*$p < 0.05$, **$p < 0.01$, ***$p < 0.001$).

the expression of six ORs genes in qRT-PCR analyses was consistent with the results of our transcriptome (Fig. 4). The *A. grahami* OF05347 and BGI_novel_G000488 were highly expressed in male, while the rest of four genes were significantly expressed in female. Among

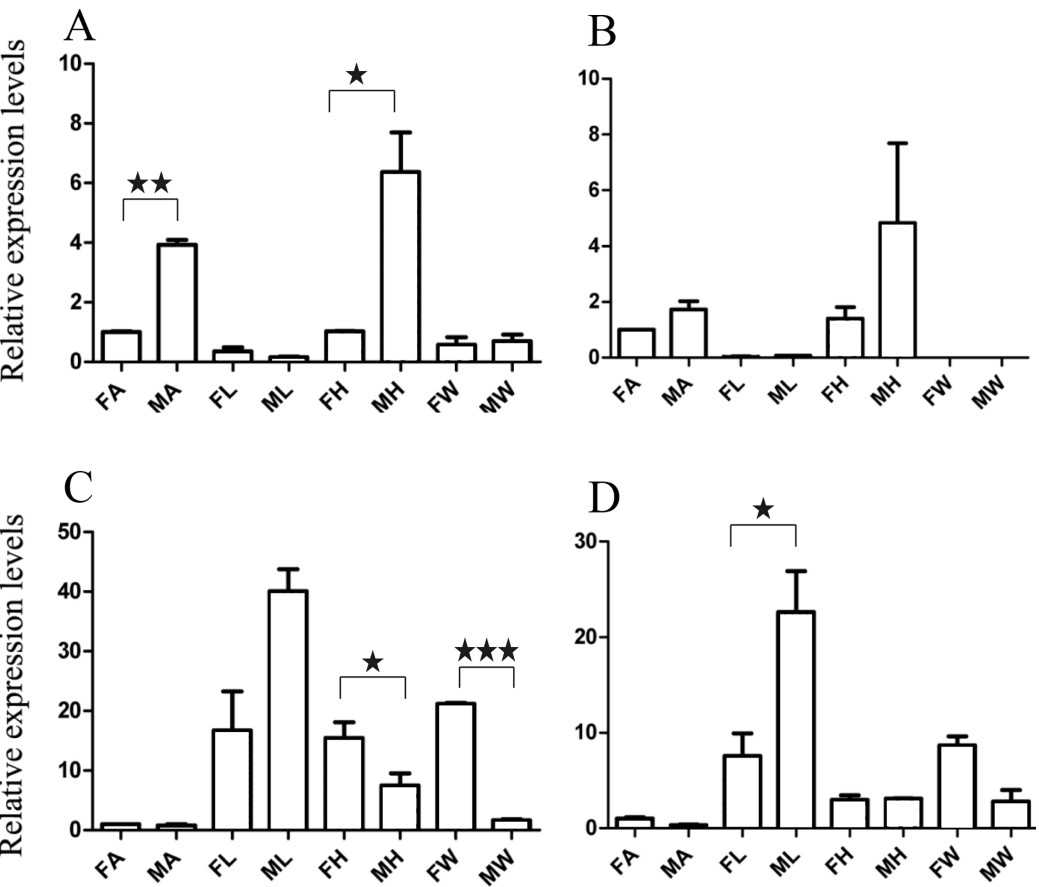

**Figure 5 Relative expression levels of *A. grahami* non-Olfactory genes.** (A) *A. grahami* OF12624. (B) *A. grahami* OF04198. (C) *A. grahami* BGI_novel_G000012. (D) *A. grahami* BGI_novel_G000010. FA, female antennae; MA, male antennae; FL, female leg; ML, male leg; FH, female head; MH, male head; FW, female wing; MW, male wing. The error bar represents standard error and the different "*" above each bar indicate significant differences in transcript abundances (*$p < 0.05$, **$p < 0.01$, ***$p < 0.001$).

these genes, OF05347 and OF11636 both exhibited gender specific expression. Particularly, OF05347 was highly expressed in male antennae, while *A. grahami* OF11636 was in female antennae (Fig. 4). The qRT-PCR results of four non-olfactory genes showed that only *A. grahami* OF12624 was different between female and male (Fig. 5).

## Expression of putative olfactory genes in different organs

Thirteen genes including three OBPs, six ORs and four non-olfactory genes were selected to explore the level of expression in olfactory and non-olfactory organs (head, leg and wing) in both females and males *A. grahami*. Three OBPs were highly expressed in head than other organs (Fig. 3). The OF08934 was highly expressed in both female head and in male antennae both. This may be due to the relative high expression of OF08934 in the female's mouth parts.

In addition, the six ORs (OF05347, OF11636, OF03270, OF03271, OF12341 and BGI-novel-G000488) genes were significantly upregulated in antennae, compared with other

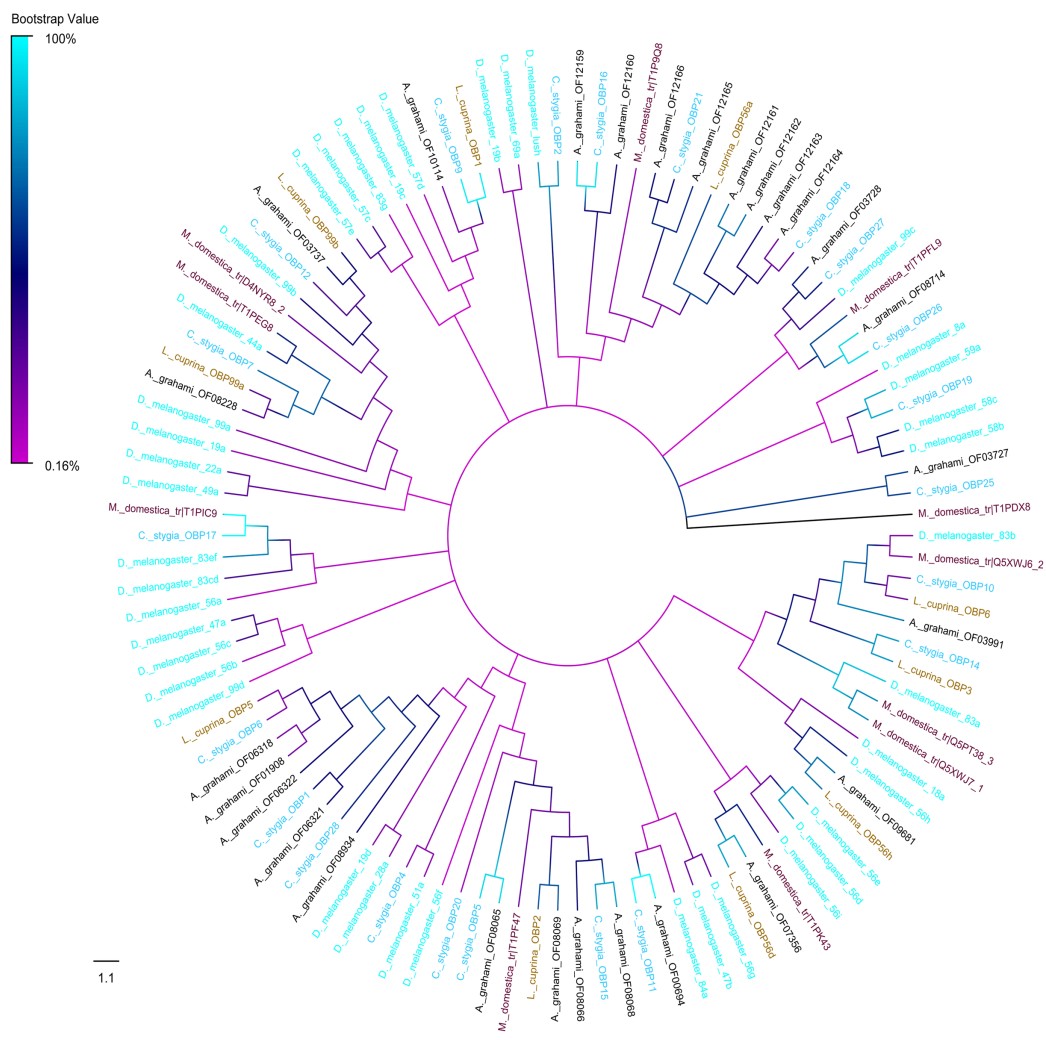

**Figure 6 Phylogenetic tree of putative OBPs from *A. grahami* and other insects.** *D. melanogaster*: *Drosophila melanogaster* (sky-blue); *C. stygia*: *Calliphora stygia* (lake blue); *M. domestica*: *Musca domestica* (purple); *L. cuprina*: *Lucilia cuprina* (yellow); *A. grahami*: *Aldrichina grahami* (black).

organs (Fig. 4). Among these genes, both OF05347 and OF11636 both exhibited organ- and gender-specific expression.

The three of four non-olfactory genes showed no difference in female and male antennae (Fig. 5). However, difference was found between organs as well as genders. *A. grahami* OF12624 was mainly expressed in the head and antennae of male, BGI-novel-G000012 was expressed in the head and wing of female, and BGI-novel-G000010 were expressed in the leg of females and males. However, for *A. grahami* OF04198 expression, there was no difference between male and female.

## Phylogenetic analysis of olfactory genes

A phylogenic tree of OBPs was constructed using OBPs of *A. grahami* with *D. melanogaster* (40), *C. stygia* (20), *M. domestica* (11) and *L. cuprina* (10) (Fig. 6).

Some pairs of *A. grahami* OBPs were paralogous genes, such as *A. grahami* OF12161/OF12164 and *A. grahami* OF12163/OF12164. In addition, the OBPs of *A. grahami*, *C. stygia*, and *L. cuprina* were grouped with higher homology, for example, *A. grahami* OF00694/*C. stygia* OBP11, *A. grahami* OF08066/*C. stygia* OBP5, *A. grahami* OF08068/*C. stygia* OBP15, *A. grahami* OF08069/*L. cuprina* OBP2 and so on. Moreover, *A. grahami* OF12159, OF12160, OF12166, OF12165, OF12161, OF12162, OF12163 and OF12164 was clustered with *D. melanogaster* LUSH (an OBP with a combination of pheromone). *A. grahami* OF03991 was clustered with *D. melanogaster* 83a/83b (a class of OBPs that were co-expressed with LUSH and related to pheromone detection), and OF06321 with *D. melanogaster* 28a.

The phylogenetic tree of *A. grahami* ORs with *D. melanogaster*, *C. stygia*, *M. domestica* and *L. cuprina* showed that ORs were clustered into multiple groups. But the *A. grahami* OF07668 was clustered into the type of ORCo. All ORCos were grouped with high support value (99%) (Fig. 7).

For IRs analysis, *A. grahami* OF11318 was clustered into a group of IR8a/25a (Fig. S3). The IR8a/25a has been proved to be IR co-receptors in other species (*Abuin et al., 2011*; *Benton et al., 2009*). *A. grahami* OF09221 was clustered into "antennal IRs" with high supporting value. Specifically, *A. grahami* OF00615 belongs to "divergent IRs", a family of proteins involved in taste detection and evaluation of ingested food before entering the digestive system (*Croset et al., 2010*). Moreover, IR25a and IR8a fell into a conserved branch in the phylogenetic tree, which was consistent with the results obtained in *Anoplophora glabripennis* (*Hu et al., 2016*).

The CSPs of three different species had relatively high conservativeness. According to the motif analysis, CSPs were found to be an extremely conservative proteins family (*Zhao et al., 2018*) (Fig. S4). The two SNMPs found in the present study exhibited high conversation with the *A. grahami* OF05479 clustering into SNMP2 (Fig. S5).

# DISCUSSION

The PMImin estimation is a major task of forensic investigation and research. But the period (pre-CI) between the point of death happened and the beginning of PMImin represent a non-negligible factor which may greatly influence the PMImin estimation. Exploration on the mechanism beneath the corpse locating behavior of necrophagous insect should facilitate the understanding of pre-CI, which will definitely improve the estimation of PMI. It has been well demonstrated that insects' olfactory system plays a major role in their food foraging behavior. Related genes and their products like OBPs, ORs, CSPs, IRs and SNMPs, were responsible for the odor detection and signal transduction. *A. grahami* is usually one of the first arrived insect group on corpses. Moreover, its obvious cold tolerance and seasonal distribution pattern could be applied as a potential "season stamp" of the time of death in the PMI estimation, especially in the period when other insects are inactive (*Kurahashi et al., 1984*; *Kurahashi, Kawai & Shudo, 1991*). Therefore, the exploration of olfactory genes and their expression feature of *A. grahami* could provide candidate genes for further research on mechanism of corpse location and factors determining the time length before insect colonization.

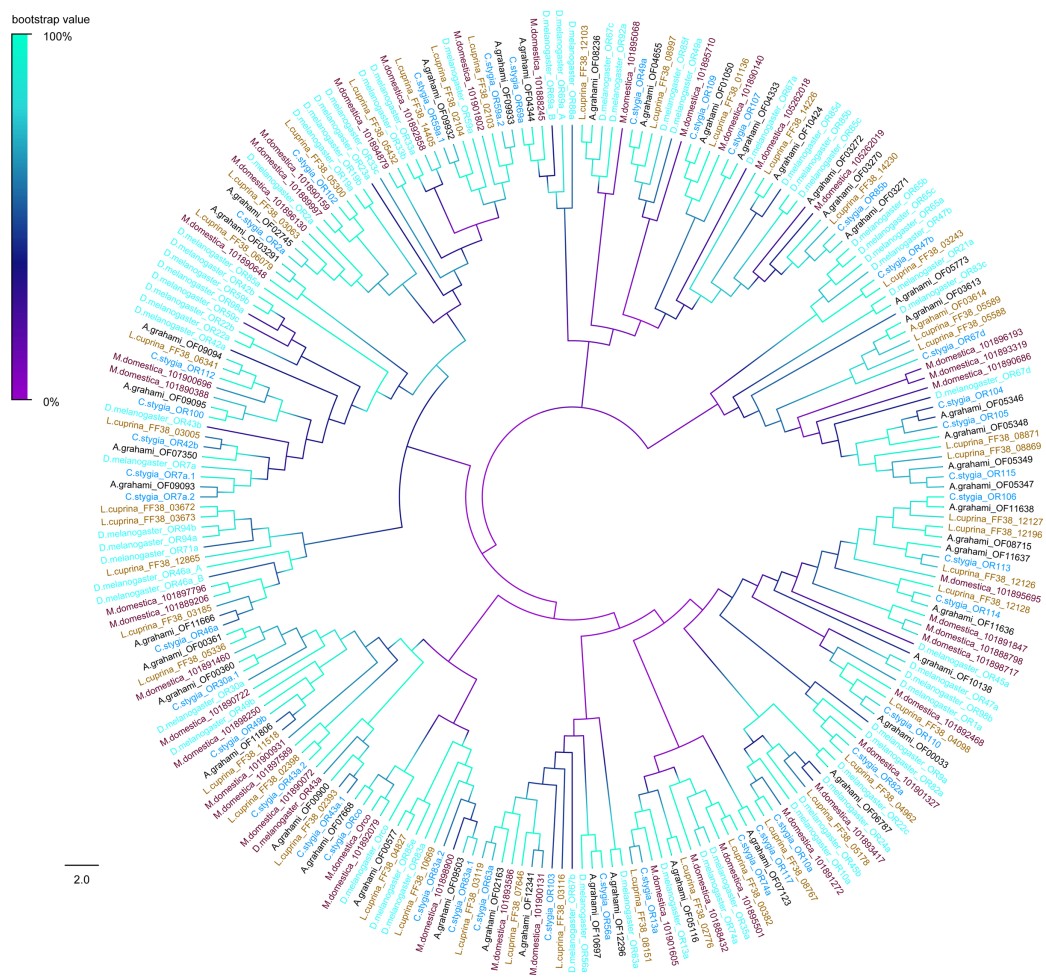

**Figure 7 Phylogenetic tree of putative ORs from *A.grahami* and other insects.** *D. melanogaster*: *Drosophila melanogaster* (sky-blue); *C. stygia*: *Calliphora stygia* (lake blue); *M. domestica*: *Musca domestica* (purple); *L. cuprina*: *Lucilia cuprina* (yellow); *A. grahami*: *Aldrichina grahami* (black).

The predicted function of genes in antennal transcriptome of *A. grahami* was found similar to other invertebrates (*Glaser et al., 2013*; *Olafson, Lohmeyer & Dowd, 2010*) (Fig. 1), especially the function of binding is the most enrich term. In general the number of olfactory gene family candidate transcripts were consistent with that of other species (*Liu et al., 2012*; *Pitts et al., 2011*; *Riveron, Boto & Alcorta, 2013*), except for CSP and IR. We also found that the expression difference of olfactory genes between the sexes is consistent to that of other insects (*Leitch et al., 2015*; *Li et al., 2018*; *Yuan et al., 2019*; *Zhao et al., 2018*). In addition, some olfactory genes are highly expressed in other organs (head, leg or wing) rather than antennae. In fact, this is common in other research results (*Li et al., 2018*; *Zhao et al., 2018*).

The premier step in pheromone or odorant perception is their interaction with members of the OBP and the CSP families (*Vieira & Rozas, 2011*). The OBPs and CSPs mainly transport odor molecules and bind to receptors on dendrites through lymphatics,

and served as a link between the external environment stimuli and odor receptor. In *A. grahami* antennae, three OBPs namely, OF08228, OF03173 and OF08934 were sex-specifically expressed in male. Interestingly, these three genes were also highly expressed in other organs compared to antennae, indicating that those genes were not solely involved in olfactory transduction. In fact, study had reported that part of OBPs could function in the gustatory system of the insect (*Jeong et al., 2013*). Additionally, two PBPs were also male-enriched. Those PBPs may play an essential function in male's perception to sex pheromone components emitted from female according to previous researches (*Allen & Wanner, 2011*; *Gu et al., 2013*). A study of *D. melanogaster* has reported that a single OBP could bind with different odors (*Liu et al., 2017*). For example, *D. melanogaster* OBP LUSH was sensitive to alcohols and 11-cis vaccenyl acetate which were found in both the active and the advanced decay stages of corpse (*Paczkowski et al., 2015*). The eight OBPs of *A. grahami* clustering with LUSH on the topological tree (Fig. 6) suggested their similar function and sensitiveness to alcohols and esters. Specifically, *A. grahami* OF00694 and *D. melanogaster* 84a were clustered into the same branch in topological tree. The latter were located in the coeloconic sensilla and functioned in detection of organic acids and amines (*Larter, Sun & Carlson, 2016*). Since organic acids and amines are vital voltaic compound during corpse decomposition, and mainly appear in the active and the advanced decay stages. These OBPs genes may play a key role in the odor preference of *A. grahami* to corpse. Although further studies are needed for identification of the specific types of odor molecules and how they may affect the blowfly.

Odorant receptors and OBPs are essential in the response of insect receptors to odors (*Barbosa, Oliveira & Roque, 2018*). In our study, 48 transcripts were typical ORs and one (*A. grahami* OF07668) was the atypical co-receptor ORCo. ORCo was previous reported highly conserved between species (*Butterwick et al., 2018*; *Jones et al., 2005*), indicating that *A. grahami* OF07668 should be subdivided as ORCo. Notably, ORCo is an important component in the regulation of insect smell, which indicates the position of other traditional ORs in olfactory sensory neurons (OSNs) of membranes (*Liu et al., 2015*). In the present study, we found gender differences in the expression level of some ORs. On the one hand, this could be explained by male and female difference in sensitiveness to similar odor stimuli. On the other hand, these differently expressed ORs may be involved in both sex pheromone and food resource detection. Based on previous a study on ORs (*Leitch et al., 2018*), different types of ORs have diverse responses to odors. Functional analysis of *D. melanogaster* odorant responses demonstrated their role in the detection of alcohol, phenols and esters, all of which are emitted during biological decomposition (*Munch & Galizia, 2016*). Based on previous research led by (*Forbes et al., 2014*), alcohols were the main compounds volatilized from corpses in winter. Some other studies suggested that mono-alcohol was attractive to flies (*Frederickx et al., 2012*; *Paczkowski et al., 2012*). The ORs of *A. grahami* identified in our study were clustered with that of *D. melanogaster* in the phylogenetic tree. For instance, *A. grahami* OF00900 and *D. melanogaster* OR43a had high supporting value and clustered together. *D. melanogaster* OR43a had a sharp response to the 1-hexanol (*Munch & Galizia, 2016*).

Furthermore, it has been reported that 1-hexanol was the component of VOCs during decomposition (*Paczkowski et al., 2015*). The former could be an important receptor of *A. grahami* in the detection of VOCs of corpses and affect foraging behavior, such as locating the corpses.

Ionotropic receptors are also transmembrane proteins which comprise the ion channels in olfactory reaction (*Gu et al., 2019*). IRs of *D. melanogaster* have been proved useful in the detection of amines and acids (*Benton et al., 2009*; *Yao, Ignell & Carlson, 2005*). In present study, there was no significant difference in the expression of six IRs between females and males. Although, one candidate IR was homologous to the antennal IRs of *D. melanogaster* (Fig. S3) was sensitive to individual decomposition compounds (e.g., propionic acid, ammonia, butyric acid and putrescine) between male and female (*Rytz, Croset & Benton, 2013*). Antennal IRs represent the basis for an accurate odor response from antenna neuron subgroup (*Croset et al., 2010*). We found that *A. grahami* OF09221 was homologous to *D. melanogaster* IR64a. Since the later gene was sensitive to acid-sensing (*Ai et al., 2010*) and organic acids, both of which are components of VOCs. *A. grahami* OF09221 is likely to play an acid-sensing role in the process of detecting corpses. Additionally, *A. grahami* OF11318 was rooted in the IR8a/25a in phylogenetic tree suggesting that it should be an ancestor the conserved member of IRs family. Besides the antennal IRs and the co-expressed receptor IRs mentioned above, the divergent IRs represent a large part of total IRs revealed by the present study. These divergent IRs are a type of proteins expressed in gustatory organs as taste receptors (*Croset et al., 2010*).

Chemosensory proteins are a group of soluble carrier proteins harboring a similar function to that of OBPs (*Wanner et al., 2004*). We identified a single putative CSP transcript and CSP protein in *A. grahami*. Therefore, this protein is less present in *A. grahami* versus other species (*Wanner et al., 2004*; *Zhang et al., 2015b*; *Zhou et al., 2010*). The reason for this difference is still unclear. The expression level, organ or species specificity were all possible.

In general, SNMPs are conversed in insects with limit family members (*Qiu et al., 2018*). Similar to previous studies, we identified two SNMPs (*Liu et al., 2013*, *2014*). Based on the phylogenetic analysis, the differently expressed *A. grahami* OF05479 is clustered into SNMP2, which is a protein that considered to be an important component of protecting olfactory function (*Blankenburg, Cassau & Krieger, 2019*) (Fig. S5). In addition, we have found that *A. grahami* OF07379 and *L. cuprina* SNMP3 were gathered in the same cluster, indicating that they possibly have a similar function. And, there are no reporters about SNMP3 in present. However, *L. cuprina* SNMP3 has high homology with SNMP1, which is a protein necessary for pheromone detection in other species (*Leitch et al., 2015*; *Rogers et al., 1997*).

In addition to the conventional protein family mentioned above, nine non-olfactory genes possibly related to olfactory transduction have also been revealed by our transcriptome. Four of the nine genes (*A. grahami* BGI_novel_G000010, BGI_novel_G000012, OF04198 and OF12624) were differently expressed when male and female antennae were compared. Based on the annotation results, these non-olfactory genes could be participant in both sensory system and signal transduction in *A. grahami* olfactory system (Table S6).

## CONCLUSION

Our current investigation is the first comprehensive analysis of antennal transcriptome from the forensically important insect, *A. grahami*. Particularly, we determined gender and organ-specifically expressed olfactory genes of *A. grahami* and discussed their potential functions in necrophagous behavior. Further studies should focus on the co-relationship between specific genes and VOCs components emitted by the corpse, as well as one the functional differences of particular olfactory gene from various necrophagous insects.

In conclusion, olfactory genes found in present paper should provide important information that can be used in the future for functional studies of *A. grahami* olfactory-associated genes and other forensic related flies. Moreover, our findings will facilitate the exploration of olfactory mechanisms in necrophagous blowfly species as well as improve PMImin estimation during forensic investigations.

### Funding

This work was supported by the National Natural Science Foundation of China (grant number 81571855) and the Science Foundation of Human Province (2017SK2015). The funders had no role in study design, data collection and analysis, decision to publish, or preparation of the manuscript.

### Grant Disclosures

The following grant information was disclosed by the authors:
National Natural Science Foundation of China: 81571855.
Science Foundation of Human Province: 2017SK2015.

### Competing Interests

The authors declare that they have no competing interests.

### Author Contributions

- Han Han conceived and designed the experiments, performed the experiments, analyzed the data, prepared figures and/or tables, authored or reviewed drafts of the paper, and approved the final draft.
- Zhuoying Liu conceived and designed the experiments, performed the experiments, analyzed the data, prepared figures and/or tables, authored or reviewed drafts of the paper, and approved the final draft.
- Fanming Meng performed the experiments, prepared figures and/or tables, authored or reviewed drafts of the paper, and approved the final draft.
- Yangshuai Jiang analyzed the data, prepared figures and/or tables, and approved the final draft.
- Jifeng Cai conceived and designed the experiments, analyzed the data, authored or reviewed drafts of the paper, and approved the final draft.

## Data Availability

The aldrichina grahami genome sequence data is available at NCBI: PRJNA513084.

The aldrichina grahami transcriptome sequence data is available at NCBI: PRJNA577237.

## Supplemental Information

Supplemental information for this article can be found online at http://dx.doi.org/10.7717/peerj.9581#supplemental-information.

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
