# Peer review of "Identification of olfactory genes of a forensically important blow fly, Aldrichina grahami (Diptera: Calliphoridae)"

_PeerJ, doi:10.7717/peerj.9581_

## Round 0.1 · original submission · Major Revisions

I think that both reviewers are in general positive about your findings and value the manuscript. I encoaurege you to prepare a new version and send us a point-by-point response to all reviewer's concerns.

Reviewer 1 ·

Basic reporting

There are sections (discussion and conclusions) that are not at the expected level of english and should be reviewed.
REferences are provided however the authors often use references that discuss previous work - the authors should use the original reference in these intances - the authors need to check many of their references for this.

Experimental design

Authors should provide further context on the behaviour and ecology of this organism to more fully explain how understanding the oflactory system will make a meaningful contribution to the forensic investigations.

Validity of the findings

Discussion and conclusions is very speculative and not particularly easy to understand. More general statements should be made and the english needs to be improved

There are far too many figures in the main body of the text - authors should include some in the supplementary information and only keep the most relevant ones that are discussed in more depth.

Additional comments

Overall i think the authors need to provide more context about the organism and its behaviour rather than rehash already well known and reviewed information about insect chemosensory systems.
I have provided specific comments below - these are not comprehensive and the authors should have the manuscript edited by a native english speaker.

Abstract
What impact will having the transcriptome and genes identified have in the field of research?
Line 20 – add ‘an’ between important role

Intro
Line 62-62 – what is meant by the behaviours are related to gender in insects? What is known about behaviours with gender differences in A. grahami??
More info on the ecology and behaviours of A. grahami would provide better context for this paper. Many many papers summarise the olfactory system of insects. Consider reducing this part and increasing the info provided about the organism you are studying instead.
Line 66 – tentacles and lower lip whiskers are not insect olfactory tissues!!!
Line 68 – the collective noun for sensillum is sensilla not sensillums
Line 76-78 – this sentence is a bit clunky, consider rewording
Line 95 – remove ‘other’

Results
Line 257 – ‘9 genes’ – what genes are these?
Line 260- 271 upregulated in what? Female?
Line 270 – how did you verify the expression – perhaps state this in this sentence
Line 279 – what does ‘were screened out’ mean? This section is difficult to understand
Line 305 – 306 what does ‘relative domain for further analysis’ mean?
Line 310 – this is unclear – are you saying there are 2 orco genes in your transcriptome? This is highly unusual
Line 336-337 – this is unclear – it reads like the male specific genes are enriched in female heads – rewrite this sentence in the context of the previous sentence.
Line 338-345 – the English is a little clunky in this section – please review
Line 346 – what do the authors mean by olfactory transduction genes? Fig 11 is olfactory receptors??

Discussion
Overall discussion needs to be edited by an English speaker
Line 351 – discussion does not need to be plural
Line 353 – ‘food resources are in distant’ – grammatically not correct please revise.
Line 360 – ‘through the receptors on lymphatics’ – re word as this is incorrect
Line 364-365 – this is the first mention of spawning – how may these olfactory differences affect location of spawning?
Line 365- 370 – this is clunky – please review the grammar
Line 370 – PBPs probably – this is too strong a statement – use ‘may’ instead of probably
Line 374 – grammar – replace sensitivity with sensitive
Line 375 – why is this relevant here? Do you mean in the advanced decay stage of corpses? This whole section needs to be much clearer
Line 382 – “these genes identify odours…’ this makes no sense
Line 386 – CSPs are not a ‘recent’ discovery
Line 388 – what evidence do the authors have the CSP is ‘less expressed’? You cannot compare the expression levels between transcriptomes of other species.
Line 389-390 – there is no developmental stage data presented in this manuscript???
Line 398 – 399 – how does A. grahami sexually communicate with pheromones? Does the male produce a Pheromone? Why do the authors think highly expressed female ORs are involved in pheromone detection? Could they not be involved with selecting locations for egg laying???
Line 391 – 411 – this whole section is very speculative – how strong are the homologies between the genes? Perhaps the authors should be more conservative in suggesting functions for the particular genes and just point out important odours for the organism and the clusters of ORs that may detect them.

Reviewer 2 ·

Basic reporting

Basic reporting well writing and scientifically sounding, but one more round of proofreading is required checking the use of some verbs and typos and add some references (ie. Line 85: “are instead of “is” to refers to the OBPs; line 227: add reference to the procedure). In abstract, Methods section authors should explain that the used NGS and DGE to antenna, and q-PCR to compare those with other tissues. Introduction provides sufficient background, except for olfactory transduction genes (lines 35, 330, 346, Fig. 12). Authors should consider to merge Results and Discussion sections in order to avoid the repetition of findings description and also to enrich the discussion going beyond of just state that results were similar to other studies. Authors should improve the discussion providing more robust quantitative comparison.

I couldn’t access to the RAW trancriptome data using the provided link or searching by PRJNA513084 code in NCBI website.

Experimental design

In general, experimental design seems to be appropriate to the objective of the study. However, some details are required to improve this section.
Line 140: Was pork intentionally left to be infested with A. grahami? How long?
Line 141: Were adult individuals kept separately? Were they virgin or mated when antennae were collected?
Line 145: How many antennae from how many individuals did the authors collect to extract the total RNA?
Line 209: What was the criteria to choose rp-49 as reference gene?
Line 210: Did the authors collect 50 antennae per replicate? Did the authors use the same insects to extract RNA sample to perform the NGS sequencing than RNA to perform qRT-PCR?
Line 212: Considering the explanation about the procedures used to obtain the antennal cDNA, do the authors use the same procedure to obtain them from other tissues? Also is important to know if heads and legs included palpi and arolium respectively, because they are well known for having chemo-sensitive functions in many insect species.

Validity of the findings

Conclusions are too general and more details are required

Additional comments

No comments

---

## Round 0.2 · Minor Revisions

Now we have got the reviews of the revised version. Reviewers appreciate (as I do too) the effort made to improve the manuscript and the good job done in the response letter as well as in the changes and comments included. As you can see below (Reviewer 2), only a few points need to be addressed for a final decision.

Reviewer 2 ·

Basic reporting

the new version of the paper has improved notoriously respect to the original one. It looks better funded, it is more pleasant to read and it is more scientifically sounding. However, still remain some issues that I think author should solve easily.
A second round of English writing checking is suggested because typos were found in the document (i.e. “conclisions” instead of “conclusions”), as well as authors have to check some reiterations of naming the same reference in a single paragraph, first as part of the sentence and then as the reference itself; i.e.: "According to the study of SSSS et al. (SSSS et al 2015)".
Figures and tables were rearranged. Many of them are now part of the supplementary information making the work more clear and concise. However I noted that some figures given in .pdf and .png format don't show similar quality. Also I suggest to improve Figures 5 and 6. Please use a more contrasting color combination in order to makes more notorious differentiation of cluster bootstrap values (such as FigS3 does).
Results are concordant with intended purpose of the work, but please note that ORCo is used to abbreviate Odorant Receptor Co-receptor instead of ORco.
As reviewer I was unable to check the raw data because link sent by authors carries to an NCBI page that note the data are still protected form public viewing. I suggest to the authors upload a version of the information to an open repository such as Figshare, just to let the editor and reviewers to check the information.

Experimental design

No comment

Validity of the findings

In discussion section: Please, beware about “suggest” that some transcripts could detect or could be sensitive to certain chemical compound just because they clustered together with a given reported sequence in a topological analysis. First of all, authors selected and cured an arbitrary group of sequences coming from certain phylogenetically related species before to perform that analysis. In second place, the actual expression and translation of some olfactory and gustatory protein is very dynamic, and the different type of proteins participate as part of a chain of events where OBPs are probably the less selective. Moreover, rearing conditions (food) didn’t emulate field condition (corpse) , so further studies are a must before to suggest transcript-chemical

Additional comments

I have to note that I tried to check the document in two different computers, having the same problems to match the line numbers listed in the rebuttal letter with the line numbers in the version of the document containing the tracked-changes. I also noted that .docx missed many line numbers across the document. Probably it was me, but unfortunately that situation made the reviewing task much more complicated in order to check the modifications and review the entire work.

---

## Round 0.3 · accepted · Accept

The last small points were fulfilled and the manuscript is acceptable. Good job!